# Small hiatal hernia and postprandial reflux after vertical sleeve gastrectomy: A multiethnic Asian cohort

Tiffany Jian Ying Lye[1]☉*, Kiat Rui Ng[2]☉, Alexander Wei En Tan[2]‡, Nicholas Syn[3]‡, Shi Min Woo[3]‡, Eugene Kee Wee Lim[1], Alvin Kim Hock Eng[1], Weng Hoong Chan[1], Jeremy Tian Hui Tan[1], Chin Hong Lim[1]☉

**1** Division of Surgery, Department of Upper Gastrointestinal & Bariatric Surgery, Singapore General Hospital, Singapore, Singapore, **2** Sheffield Medical School, University of Sheffield, Sheffield, United Kingdom, **3** Yong Loo Lin School of Medicine, National University of Singapore, Singapore, Singapore

☉ These authors contributed equally to this work.
‡ These authors also contributed equally to this work.
* tiffany.lye.j.y@singhealth.com.sg

## Abstract

### Background

Laparoscopic vertical sleeve gastrectomy (LSG) is a popular bariatric procedure performed in Asia, as obesity continues to be on the rise in our population. A major problem faced is the development of de novo gastroesophageal reflux disease (GERD) after LSG, which can be chronic and debilitating. In this study, we aim to assess the relationship between the presence of small hiatal hernia (HH) and the development of postoperative GERD, as well as to explore the correlation between GERD symptoms after LSG and timing of meals. In doing so, we hope to gain a better understanding about the type of reflux that occurs after LSG and take a step closer towards effectively managing this difficult to treat condition.

### Methods

We retrospectively reviewed data collected from patients who underwent LSG in our hospital from Dec 2008 to Dec 2016. All patients underwent preoperative upper GI endoscopy, during which the identification of hiatal hernia takes place. Patients' information and reflux symptoms are recorded using standardized questionnaires, which are administered preoperatively, and again during postoperative follow up visits.

### Results

Of the 255 patients, 125 patients (74%) developed de novo GERD within 6 months post-sleeve gastrectomy. The rate of de novo GERD was 57.1% in the group with HH, and 76.4% in the group without HH. Adjusted analysis showed no significant association between HH and GERD (RR = 0.682; 95% CI 0.419 to 1.111; P = 0.125). 88% of the patients who developed postoperative GERD reported postprandial symptoms occurring only after meals, and the remaining 12% of patients reported no correlation between the timing of GERD symptoms and meals.

**Data Availability Statement:** All relevant data are within the manuscript and its Supporting Information files.

**Funding:** The authors received no specific funding for this work.

**Competing interests:** The authors have declared that no competing interests exist.

## Conclusion

There is no direct correlation between the presence of small hiatal hernia and GERD symptoms after LSG. Hence, the presence of a small sliding hiatal hernia should not be exclusion for sleeve gastrectomy. Electing not to perform concomitant hiatal hernia repair also does not appear to result in higher rates of postoperative or de novo GERD.

## Introduction

With obesity on the rise in Asia, bariatric surgery has also gained popularity, although the number and type of bariatric procedures performed varies significantly between countries. Laparoscopic sleeve gastrectomy (LSG) has gained popularity because of its ease, speed and safety. It is currently the most frequently performed bariatric surgery procedure in the Asia Pacific region, accounting for >50% of all bariatric procedures [1].

Compared to laparoscopic roux-en-Y gastric bypass, LSG has the advantages of relative technical simplicity with fewer long-term serious postoperative complications, and similar outcomes in terms of weight loss. However, there has been increasing concern regarding the prevalence of gastro-esophageal reflux disease (GERD) after LSG [2–5]. Although pre-operatively existing GERD might be improved after LSG due to successful weight loss and decrease in intra-abdominal pressure, yet many patients develop de novo GERD or worsening of their pre-existing reflux symptoms.

A hiatus hernia is a known independent risk factor for the development of GERD, and its prevalence is higher in obese individuals. The presence of hiatal hernia increases the distance between LES and diaphragmatic crus which may defect the anti-reflux mechanism and lead to the development of GERD. Furthermore, obese individuals have increased intra-abdominal and intra-gastric pressure, and thus a favourable gastro-esophageal pressure gradient for reflux. Suter et al. [6] found that the rate of symptomatic reflux in morbidly obese patients was 35.8%, out of which 53% had HH. Wilson et al. [7] demonstrated an association between excess weight, HH and reflux esophagitis, thus recommending the need for preoperative assessment. Nevertheless, the literature presents conflicting results concerning the effects of LSG on GERD in patients with HH [8]. Also, the effectiveness of concurrent hiatal hernia repair in reducing postoperative reflux symptoms after LSG is unclear.

In this study, our primary aim is to explore the relationship between the presence of small sliding hiatal hernia and postoperative GERD in patients who undergo LSG in our local Asian population. This can help to provide us with insight on whether or not concurrent hiatal hernia repair would be beneficial for this group of patients. Our secondary aim is to assess the correlation between GERD symptoms after LSG with the timing of meals. In doing so, we hope to better understand the mechanisms involved in GERD.

## Methods

We identified all patients who underwent LSG in our institution from December 2008 to December 2016. Patients were considered for bariatric surgery if they had body mass index (BMI) >37.5 kg/m2 or BMI >32.5kg/m2 with obesity-related comorbidities [9]. All patients were pre-operatively evaluated by a multidisciplinary team consisting of dietitians, endocrinologists, physiotherapists, psychologists and bariatric surgeons at our weight management program.

Pre-operatively, the patients' baseline demographics and anthropometrics were recorded. Upper GI endoscopy was performed for all patients before surgery. The diagnosis of hiatal hernia was made based on the presence of a diaphragmatic indentation of at least 2 cm distal to the squamocolumnar junction or Z-line and the proximal margins of the gastric mucosal folds on endoscopic examination (Fig 1) [10]. Since movement of the GEJ within the range of 2 cm occurs during normal swallowing and is considered physiologic, it is commonly believed that sliding hiatal hernia to exceed this range should be considered clinically significant [11].

Antibiotic prophylaxis was administered in compliance with our bariatric institutional protocol. All LSGs were performed using five ports. Beginning 3–5 cm proximal to the pylorus, the omentum was separated from the greater curvature by dividing the branches of the gastro-epiploic vessels and the short gastric vessels just adjacent to the stomach serosa. The fundus was fully mobilized, exposing the left crus in all cases. Care was taken to transect the fundus off the sleeve approximately 1 cm lateral to the angle of His to avoid placing the most proximal staple line at the gastroesophageal junction. A 38-Fr calibration tube was used to size the gastric tube before division of the stomach. With the calibration tube in situ, longitudinal division of the stomach was accomplished by consecutive applications of an endoscopic stapler from 3–5 cm proximal to the pylorus to the gastroesophageal junction. We do not routinely inspect or repair small hiatal hernias because we believe aggressive interrogation of the hiatus may lead to disruption of the integrity of the sling fibers of Helvetius at the esophagogastric junction, thus contributing to the incidence of new or worsening postoperative GERD. We defined small hiatal hernia as sliding hernia >2 cm but <5 cm from the squamocolumnar junction during upper GI endoscopy. Patients with large sliding hiatal hernia >5cm and those with paraesophageal hernia underwent concurrent hiatal hernia repair and this group of patients were excluded from this study.

On the first postoperative day, all patients were commenced on our post-bariatric surgery protocol, which included small quantities of clear liquids, progressing to a full liquid diet by the afternoon. Discharge of the patient usually occurred on postoperative day 1–2 once discharge criteria were achieved. Patients were reviewed by our multidisciplinary team 2 weeks postoperatively, followed by review at 1 month, 3 months, 6 months and annually thereafter.

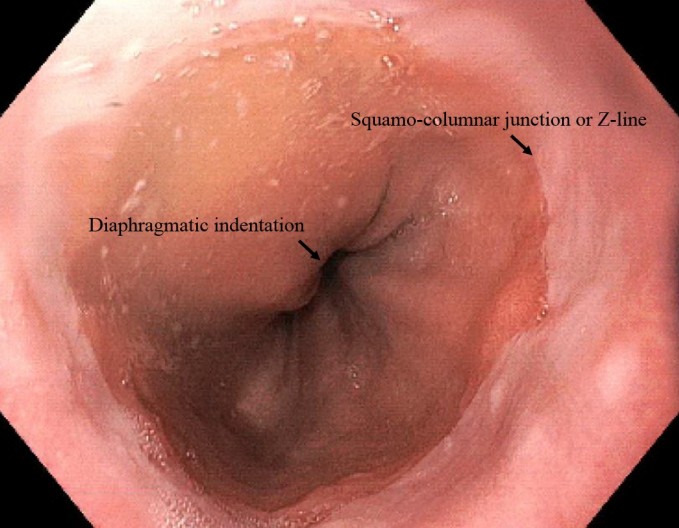

**Fig 1. Endoscopic diagnosis of a hiatus hernia.** Hiatal hernia diagnosis is made based on the presence of a diaphragmatic indentation of at least 2 cm distal to the squamo-columnar junction or Z-line.

Discharge criteria included (1) drinking 1.5 L of fluid per day and tolerating the prescribed liquid diet, (2) pain adequately controlled with oral analgesia, (3) ability to ambulate without assistance and (4) understanding and acceptance of the written information sheets provided.

Our bariatric surgery unit's standardized questionnaire (Fig 2) was administered to all patients preoperatively and at subsequent postoperative follow-up visits. The questionnaires are aimed at assessment of reflux symptoms, and also includes other relevant information such as patients' smoking and alcohol history, and use of acid suppression medication. Patients who are lost to follow-up and those with incomplete data were contacted via phone survey. Patients were categorized pre-operatively as having GERD or no GERD. Postoperatively, patients rated responses as resolution/ improvement of GERD symptoms or remain unchanged or worse. Post-operatively, upper GI endoscopy is not routinely performed unless patients complain of GERD symptoms.

The study was approved by our institution's review board (Singhealth Centralized IRB). Verbal consent was obtained from all patients included in this study. The patients' data were obtained from hospital medical records and from their individual questionnaires.

Collected data were analyzed with Prism version 6 software (GraphPad Software, Inc., La Jolla, CA). Descriptive results regarding categorical variables were given as percentages (%) of subjects affected. Normally distributed continuous variables are presented as the mean ± standard deviation (SD). Following LSG, patients were divided into two groups according to the presence or absence of HH. Comparisons of independent variables were done via Student's unpaired t-tests. Chi-square tests were performed for categorical variables. A p value of $<0.05$ was considered statistically significant. In addition, Poisson generalized linear models with a log-link function and robust (sandwich) error variance were used in our statistical analysis. To minimize confounding, models were adjusted for patient's height, previous use of proton-pump inhibitors or antacids, smoking and alcohol use, as well as for interaction terms between smoking or alcohol use and prescription gastric acid suppressants.

## Results

From December 2008 to December 2016, a total of 417 obese patients underwent LSG at our hospital (Fig 3). Twenty-four patients (5.8%) were subsequently excluded due to either the lack

| History | Never | Sometimes | Always |
|---|---|---|---|
| 1. Do you have burning sensation or burning pain in your stomach or behind your breastbone (heartburn)? | | | |
| 2. Do you have stomach content moving upwards to your throat or mouth? | | | |
| 3. Do you experience belching or bloating? | | | |
| 4. Does it happen within first 2 h after eating? | | | |
| 5. Does it happen at any time and there is no relation to eating? | | | |
| 6. Does it happen only when you eat a lot or more than you are accustomed to? | | | |
| 7. Does it happen only when you eat too fast? | | | |
| 8. Does it improve with antacids or Omeprazole? | | | |
| 9. Do you smoke? | | | |
| 10. Do you drink alcohol? | | | |

**Fig 2. Standardized questionnaire for GERD in bariatric patients.**

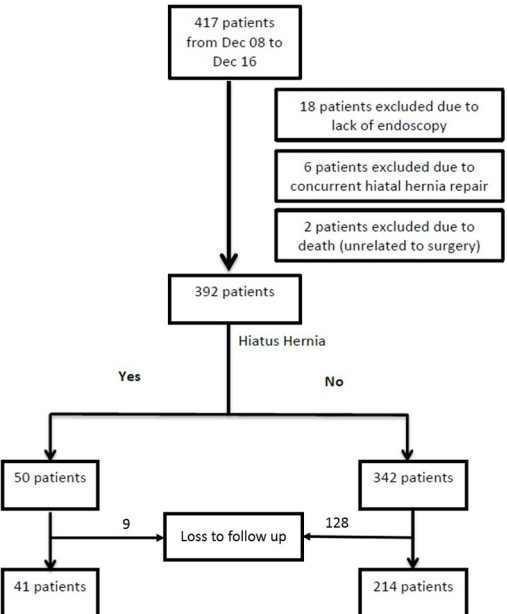

**Fig 3. Patient selection flowchart.**

of preoperative endoscopy or concurrent HH repair (patients with paraesophageal hernia types II, III and IV or sliding hernia >5 cm). There were 2 deaths secondary to malignancy. 137 patients were lost to follow up. The final study population included 255 patients. The mean age of the cohort was 41.04 ± 10.65 years, the majority of whom were female (62.7%). The group was predominantly Chinese (39.2%), with Malays being the second largest group (23.9%). There were no cases of operative mortality or conversion to open. There were no major complications such as leak, sleeve stenosis or stricture in the study group. Preoperatively, 41 (16.1%) of our patients had type I hiatal hernia upon routine upper endoscopy. The median follow-up duration was 26 months (6 months to 9 years).

The mean BMI was 43.64 ± 8.02 pre-operatively, and 31.58 ± 6.54 post-operatively. The mean percentage of excess weight loss (% EWL) was 60% for the cohort. There were no significant differences in terms of weight loss between patients with or without hiatal hernia. Percentage of excess weight loss had no impact on patient's reporting symptoms of GERD. The baseline demographics of the patients with HH and without HH are summarized in Table 1.

Preoperatively, 86 (33.7%) of the patients reported typical GERD symptoms of heartburn and regurgitation. Of these, only 25.9% required daily anti-reflux medication. Among the 169 patients who were asymptomatic before surgery, 125 patients (74%) developed de novo GERD within 6 months post-sleeve gastrectomy. The rate of de novo GERD was 57.1% in the group with HH, and 76.4% in the group without HH. Interestingly, among patients who had GERD pre-operatively, 20 patients (23.3%) experienced resolution of their GERD symptoms post-operatively (Table 2). Another 10 patients reported ongoing but improved GERD symptoms. The rest of the patients had no change or worsening of their symptoms after LSG.

In the unadjusted analyses of patients with and without HH developing GERD symptoms, surprisingly, patients with HH appeared to have a lower risk for developing reflux (relative risk [RR] = 0.611; 95% CI 0.393 to 0.949; P = 0.028). However, after analyses were adjusted for height, previous use of proton-pump inhibitors or antacids, smoking and alcohol use, as well as for interaction terms between smoking or alcohol use and prescription gastric acid

**Table 1. Patient's baseline characteristics.**

|  | **No Hiatal Hernia** | **Hiatal Hernia** | *p* |
|---|---|---|---|
|  | **n = 214** | **n = 41** |  |
| Mean age | 41.39 ±11.00 | 28.80 ±8.37 | 0.154 |
| Gender |  |  | 0.103 |
| Male, n (%) | 77 (36) | 18 (44) |  |
| Female, n (%) | 137 (64) | 23 (56) |  |
| Race |  |  | 0.151 |
| Chinese, n (%) | 87 (41) | 13 (32) |  |
| Malay, n (%) | 63 (29) | 15 (37) |  |
| Indian, n (%) | 53 (25) | 11 (27) |  |
| Other, n (%) | 11 (5) | 5 (4) |  |
| Smoking, n (%) | 46 (21) | 3 (7) | 0.306 |
| Alcohol Consumption, n (%) | 55 (26) | 2 (5) | 0.258 |
| PPI Pre-LSG | 39 (17) | 27 (21) | 0.646 |
| PPI Post-LSG | 65 (30) | 8 (20) | 0.187 |
| Preop weight | 118.2 ±1.72 | 116.7 ±3.01 | 0.362 |
| Postop weight | 85.6 ±1.33 | 83.0 ±2.25 | 0.208 |
| Actual weight loss | 32.6 ±1.02 | 33.7 ±2.79 | 0.335 |
| % EWL | 64.3 ±1.75 | 60.8 ±3.85 | 0.210 |

suppressants, there was no statistically significant association between HH and GERD (RR = 0.682; 95% CI 0.419 to 1.111; P = 0.125).

Further subgroup analysis was performed on the patients who have post-op GERD to determine the correlation between GERD symptoms and the timing of meals (Table 3). 168 (88%) of the patients reported GERD symptoms occurring after meals only, and the remaining 12% of patients reported no correlation between the timing of GERD symptoms and meals. In the patients with no HH, 86.1% of the patients with post-op reflux had postprandial reflux. In patients with HH, all of the 25 patients with post-op GERD reported postprandial reflux.

## Discussion

The main aim of this study was to determine whether the presence of small type I sliding HH detected on preoperative endoscopy would present itself as a risk factor for GERD in patients

**Table 2. Prevalence of GERD pre- and post-sleeve gastrectomy.**

| **All patients (N = 255)** | | | |
|---|---|---|---|
|  | Post-op asymptomatic | Post-op GERD | Total |
| Pre-op asymptomatic | 44 (26%) | 125 (74%) | 169 (100%) |
| Pre-op GERD | 20 (23.3%) | 66 (76.6%) | 86 (100%) |
| **HH Group (N = 41)** | | | |
|  | Post-op asymptomatic | Post-op GERD | Total |
| Pre-op asymptomatic | 9 (42.9%) | 12 (57.1%) | 21(100%) |
| Pre-op GERD | 7 (35%) | 13 (65%) | 20 (100%) |
| **Without HH Group (N = 214)** | | | |
|  | Post-op asymptomatic | Post-op GERD | Total |
| Pre-op asymptomatic | 35 (23.6%) | 113 (76.4%) | 148 (100%) |
| Pre-op GERD | 13 (19.7%) | 53 (80.3%) | 66 (100%) |

**Table 3. Subgroup analysis of patients with post-op GERD.**

|  | Total post-op GERD (N = 191) | No HH group (N = 166) | HH group (N = 25) |
|---|---|---|---|
| Postprandial reflux | 168 (88%) | 143 (86.1%) | 25 (100%) |
| All day reflux | 23 (12%) | 23 (13.9%) | 0 |

undergoing LSG. We evaluated patients with small hiatal hernia diagnosed upon upper endoscopy before undergoing laparoscopic sleeve gastrectomy alone, without concomitant hiatal hernia repair. Our results showed that the presence of the small sliding hiatal hernia itself was not a risk factor for symptomatic GERD post-sleeve gastrectomy. To our knowledge, this is the largest series to date, with a median follow-up of 26 months.

Currently, there is no algorithm available for the management of HH in patients undergoing sleeve gastrectomy. Current guidelines from the Society of American Gastrointestinal and Endoscopic Surgeons (SAGES) state that all hernias detected during the course of bariatric operation should be repaired [12]. However, the quality of evidence was weak, with conflicting results (Table 4) [13–19]. Most experts do not consider small hiatal hernia a contraindication to laparoscopic sleeve gastrectomy [20]. However, the current evidence on this topic is limited by several factors: 1) there are very few studies including more than 100 patients; 2) the mean follow-up is short; and 3) those studies that describe hiatal hernia repair reported different ways to close the hiatus: suture posterior cruroplasty, suture anterior cruroplasty and hiatal herniorrhaphy with mesh (biological or polypropylene mesh).

The prevalence of hiatal hernia in the Asian population is lower compared to Western populations. Population studies conducted in Sweden, Italy and China have shown a HH prevalence of 23.9%, 43.0% and 0.7%, respectively [21]. To further complicate the matter, the accurate diagnosis of small HH is challenging. Upper GI endoscopy is the standard tool for assessing upper GI symptoms and is part of routine preoperative work-up for bariatric surgery in Asia in view of the high prevalence of gastric malignancy. Most experts consider a hiatal hernia to be present if a diaphragmatic indentation 2 cm or more is observed distal to the Z-line and the top of the stomach mucosal folds. In majority of the studies on LSG and

**Table 4. Current evidence for concomitant sleeve gastrectomy with hiatal hernia repair.**

|  | Year | n | Study design | Study population | Follow-up (months) | Results |
|---|---|---|---|---|---|---|
| Soricelli et al | 2010 | 6 | Prospective study | Concomitant LSG & HHR | 4 | 66.6% GERD symptoms |
|  |  |  |  |  |  | 75% resolution |
| Daes et al | 2012 | 34 | Cohort study | Concomitant LSG & HHR | 6–12 | 85.3% GERD symptoms |
|  |  |  |  |  |  | 93.1% resolution |
| Soricelli et al | 2013 | 97 | Prospective study | Concomitant LSG & HHR | 18 | 42.2% GERD symptoms |
|  |  |  |  |  |  | 80.4% resolution |
| Santonicola et al | 2014 | 78 vs. 102 | Prospective controlled study | Concomitant LSG & HHR vs. LSG alone | >6 | Lower GERD symptoms with LSG alone |
| Samakar et al | 2016 | 58 | Retrospective study | Concomitant LSG & HHR | 8 | 44.8% GERD symptoms |
|  |  |  |  |  |  | 34.6% resolution |
|  |  |  |  |  |  | 65.4% persistent |
|  |  |  |  |  |  | 15.6% de novo |
| El Chaar et al | 2016 | 56 vs. 239 | Retrospective study | Concomitant LSG & HHR vs. LSG alone | NA | Decrease GERD symptoms in both group |
| Snyder et al | 2016 | 100 | Prospective randomized controlled study | Concomitant LSG & HHR vs. LSG alone | 12 | No difference |

concomitant hiatal hernia (HH) repair the presence of HH is diagnosed with pre-operative upper endoscopy. Soricelli et al described that finding a macroscopically evident fingerprint indentation of the diaphragm above the esophageal emergence from the diaphragm is considered suspicious for HH, necessitating careful exploration of the diaphragmatic crura. In several of the studies, the presence of a HH was also diagnosed intraoperatively. El Chaar et al reported routine dissection of the angle of His, taking down the phrenoesophageal ligament and mobilization of the fat pad in order to identify and measure HH, regardless of whether HH was diagnosed on upper endoscopy or not. We believe that intraoperative interrogation and dissection of the hiatus is extremely unreliable for the diagnosis of small HHs and is subject to operator discretion. This aggressive inspection has led many surgeons to open the phrenoesophageal ligament and, in a sense, create a small hernia defect that is then sutured closed more tightly. Furthermore, disruption of the integrity of the sling fibres of Helvetius at the esophagogastric junction may potentially contribute to increased reflux. The added risks of HH repair and the extra operative time and cost need to be taken into consideration when deciding to fix a small HH. Although most complications are minor, e.g., dysphagia, pneumothorax, nausea and vomiting, Chang et al. reported a case of death from haemorrhage with simultaneous sleeve gastrectomy [22]. Recent evidence has shown that the use of barium swallow X-ray provides the highest rate of HH detection [23]. Nevertheless, there is no standardized protocol regarding whether the X-ray should be done in the upright or supine position, adding to the inconsistency in diagnosing hiatal hernias. Furthermore, swallowing itself distends and shortens the esophageal lumen, making diagnosis of small hiatal hernia impractical with a barium swallow. In 2 of the studies, preoperative UGI contrast study was performed in all patients [16, 17]. However, Satonicola et al qualified that contrast study may be able to diagnose a large HH, but the diagnosing of small HH would be challenging. In the study by Samakar et al, only UGI contrast study only detected HH in 34.5% of patients.

The subgroup analysis on patients with GERD symptoms after LSG demonstrated that the majority of the patients experience postprandial reflux symptoms, with a minority experiencing all-day symptoms. We postulate that the reflux symptoms after LSG may be related to non-acid volume reflux instead of acid reflux. Reduced compliance of the gastric tube, increased intraluminal pressure when the pylorus is closed, disruption of the angle of His causing impairment of the LES antireflux mechanism, and a funnel shape of the gastric tube are among some of the proposed mechanisms that contribute to volume reflux after LSG. Furthermore, we would expect that resection of the gastric fundus would result in decreased acid production. This leads us to hypothesize that the use of acid suppressant medications to treat GERD after LSG may not be very effective, and perhaps dietary modification plays a more important role in improving their symptoms, with the consideration of revisional surgery if symptoms fail to improve.

The incidence of de novo GERD after laparoscopic sleeve gastrectomy decreases with time after surgery. Himpens et al. demonstrated in a prospective randomized study that the incidence of de novo GERD continues to decrease with time, dropping from 21.8% after 1 year, to 3.1% after 3 years from the time of surgery. It was postulated that the rationale for this could be due to a gradual increase in gastric tube compliance and gastric emptying with time. From our data, the 74% of patients that develop de novo GERD in our study reflects the total number of patients that develop new GERD symptoms at any point of time after surgery, and thus the incidence is high especially within the first 6 months, but is expected to improve with time. Our study data shows that after 6 months, there was in fact improvement of de novo GERD from 74% to 48.5%. We also postulate that eating behaviour plays an important role as the patients tend to overeat during the early post-operative period, resulting in increased intragastric pressure and higher likelihood of reflux. Dietary adjustment takes time and subsequently results in improvement or resolution of GERD symptoms.

This study has its limitations. First, endoscopic evaluation may not be the best modality for diagnosing small HH, but nevertheless it is probably superior to intraoperative assessment. Second, there was no objective evaluation of GERD postoperatively with EGD, contrast studies, 24-hour ambulatory pH monitoring or impedance studies. Although some studies do use these methods to confirm the diagnosis of GERD, they are not required as depicted by the 5th International Consensus Conference on the Current Status of Sleeve Gastrectomy [24]. Nevertheless, this paper has significant sample size and relatively longer follow-up compared to most other studies.

Although our unit's questionnaire was not validated but was more relevant to post-sleeve gastrectomy subjects, it takes into account smoking and alcohol history, symptoms related to meals, PPI usage, overeating and differentiation between acid reflux versus volume reflux [25]. We found that commonly used validated questionnaires, such as GERD-Q and GERD health-related quality of life (GERD-HRQL), may be useful in correlating symptoms to presence of esophagitis, but extremely impractical to assess progression or response of symptoms over time. Furthermore, in patients who have undergone LSG, it is thought that their symptoms are mainly related to regurgitation and volume reflux. These commonly used questionnaires are useful at identifying and assessing symptoms related to acid reflux but may not completely and adequately assess patients' symptoms which are due to volume reflux and postprandial regurgitation. Althuwani et al. had concluded that 35.7% of reflux is in fact non-acid regurgitation [26]. Another difference to highlight is that our unit's standardized questionnaire utilizes a dichotomous scale for patient responses, as opposed to other GERD questionnaires which quantify responses using a likert type ordinal scale. Cultural differences do play a role in influencing the responses in likert scales. Our multiracial and multicultural Asian study population reflects our country's patient population well, and thus opting to use a dichotomous scale in our questionnaire aims to limit response style bias. It also minimizes the possibility of recall bias given the duration between the surgery and post-operative application of the questionnaire. The use of a dichotomous scale in our questionnaire can also explain the higher rates of symptomatic GERD after LSG (74%) compared to other studies, as it takes into account all patients who have GERD symptoms but does not distinguish symptom severity.

## Conclusion

Our study demonstrates that there is no direct correlation between the presence of small HH and GERD symptoms post-LSG. Hence, the presence of a small sliding hiatal hernia should not exclude patients from having a laparoscopic sleeve gastrectomy. Electing not to perform concomitant hiatal hernia repair also does not appear to result in higher rates of postoperative or de novo GERD. We believe that further studies need to be performed to confirm the type of reflux that occurs after LSG in order for us to gain a better understanding of how to go about managing this difficult to treat condition.

## Supporting information

**S1 Dataset.**
(XLSX)

## Author Contributions

**Conceptualization:** Chin Hong Lim.

**Data curation:** Kiat Rui Ng, Alexander Wei En Tan, Nicholas Syn, Shi Min Woo.

**Formal analysis:** Tiffany Jian Ying Lye, Kiat Rui Ng, Nicholas Syn.

**Methodology:** Kiat Rui Ng, Chin Hong Lim.

**Resources:** Eugene Kee Wee Lim, Alvin Kim Hock Eng, Weng Hoong Chan, Jeremy Tian Hui Tan, Chin Hong Lim.

**Supervision:** Eugene Kee Wee Lim, Alvin Kim Hock Eng, Weng Hoong Chan, Jeremy Tian Hui Tan, Chin Hong Lim.

**Writing – original draft:** Tiffany Jian Ying Lye.

**Writing – review & editing:** Tiffany Jian Ying Lye, Chin Hong Lim.

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
