## [Decision Letter · Decision Letter 0]

26 Aug 2020

PONE-D-20-04688

Small hiatal hernia and postprandial reflux after vertical sleeve gastrectomy: a multiethnic Asian cohort

PLOS ONE

Dear Dr. Lye,

Thank you for submitting your manuscript to PLOS ONE. After careful consideration, we feel that it has merit but does not fully meet PLOS ONE’s publication criteria as it currently stands. Therefore, we invite you to submit a revised version of the manuscript that addresses the points raised during the review process.

Please reply and give appropriate response specific to the reviewers' comments. Final decision will depend on your reply. Thanks.

We look forward to receiving your revised manuscript.

Kind regards,

Chun Chieh Yeh, M.D., Ph.D.

Academic Editor

PLOS ONE

Journal Requirements:

2.We note that you have indicated that data from this study are available upon request. PLOS only allows data to be available upon request if there are legal or ethical restrictions on sharing data publicly. For information on unacceptable data access restrictions, please see http://journals.plos.org/plosone/s/data-availability#loc-unacceptable-data-access-restrictions.

Additional Editor Comments (if provided):

Sorry for delay in reply. It is challenging to find appropriate reviewers for this topic at this moment. Please reply and give appropriate response specific to the reviewers' comments. Final decision will depend on your reply. Thanks.

Reviewers' comments:

Reviewer's Responses to Questions

**Comments to the Author**

1. Is the manuscript technically sound, and do the data support the conclusions?

Reviewer #1: Yes

Reviewer #2: Yes

2. Has the statistical analysis been performed appropriately and rigorously? 

Reviewer #1: Yes

Reviewer #2: N/A

3. Have the authors made all data underlying the findings in their manuscript fully available?

Reviewer #1: No

Reviewer #2: Yes

4. Is the manuscript presented in an intelligible fashion and written in standard English?

Reviewer #1: Yes

Reviewer #2: Yes

5. Review Comments to the Author

Reviewer #1: The authors present a retrospective analysis of patients undergoing laparoscopic sleeve gastrectomy in a multi-ethnic Asian cohort, focusing on GERD outcomes as a function of the presence/absence of small hiatal hernias. The study is well written and provides a systematic assessment from preoperative evaluation in lieu of intraoperative assessment alone as several other studies have.

Study data could be anonymized and posted publicly.

Studies cited in the current evidence table should be listed in the references section.

The method of assessment for hiatal hernial in each study may be a beneficial addition for readers.

Is it know whether there was any reduction in PPI dose following surgery?

Reviewer #2: 1. Please discuss in discussion section that if 75% of patients develop de novo GERD then why continue to perform sleeve gastrectomy? That is an excessive amount of de novo GERD.

2. In method section, please state the questionnaire that was used to assess GERD preop and postop. Is the GERD questionnaire validated?

3. The methods section stated that the patient answered if GERD is resolve, improved, unchanged or worsen. However, the results section in the tables only showed if GERD is presence or not in the postoperative period. Please show details of results such as how many patients with continued GERD symptoms but "improved" compared to preop.

6. PLOS authors have the option to publish the peer review history of their article (what does this mean?). If published, this will include your full peer review and any attached files.

Reviewer #1: No

Reviewer #2: No

---

## [Author Response · Author response to Decision Letter 0]

19 Sep 2020

Dear reviewers,

We appreciate and thank you for your time spent and for the invaluable comments that will certainly help to improve the quality and robustness of our manuscript. I do hope that your questions and suggestions have been adequately addressed in the responses below, as well as in the revised manuscript.

Responses to Reviewer #1

1) Study data could be anonymized and posted publicly.

We have anonymized and submitted the study data. 

2) Studies cited in the current evidence table should be listed in the references section.

The citations have been added into the revised manuscript.

3)The method of assessment for hiatal hernia in each study may be a beneficial addition for readers.

In majority of the studies on LSG and concomitant hiatal hernia (HH) repair, most of them diagnose the presence of HH with pre-operative upper endoscopy. Soricelli et al described that finding a macroscopically evident fingerprint indentation of the diaphragm above the esophageal emergence from the diaphragm is considered suspicious for HH, necessitating careful exploration of the diaphragmatic crura. In several of the studies, the presence of a HH was also diagnosed intraoperatively. El Chaar et al reported routine dissection of the angle of His, taking down the phrenoesophageal ligament and mobilization of the fat pad in order to identify and measure HH, regardless of whether HH was diagnosed on upper endoscopy or not. In our unit we do not perform this routinely for small hiatal hernias as we believed that disruption of the integrity of the sling fibres of Helvetius at the esophagogastric junction may potentially contribute to increased reflux. In 2 studies (Satonicola, Samakar), all patients also underwent a preoperative UGI contrast study. However, Satonicola et al qualified that contrast study may be able to diagnose a large HH, but the diagnosing of small HH would be challenging. In the study by Samakar et al, only UGI contrast study only revealed HH in 34.5% of patients, and the remaining HH were diagnosed intra-operatively. 

4) Is it known whether there was any reduction in PPI dose following surgery?

Majority of the studies cited have had no data presented on changes in the dosage of PPI after surgery. Only Soricelli et al reported 80.4% of the patients had discontinuation of PPI after LSG with HH repair, and 19.6% of patients needed reduced dose of PPI (40mg/d to 15mg/d). 

Responses to Reviewer #2

1) Please discuss in the discussion section that if 75% of patients develop de novo GERD then why continue to perform sleeve gastrectomy? 

The incidence of de novo GERD after laparoscopic sleeve gastrectomy decreases with time post operatively. Himpens et al. demonstrated in a prospective randomized study that the incidence of de novo GERD continues to decrease with time, dropping from 21.8% after 1 year, to 3.1% after 3 years from the time of surgery. It was postulated that the rationale for this could be due to a gradual increase in gastric tube compliance and gastric emptying with time. From our data, the 74% of patients that develop de novo GERD in our study reflects the total number of patients that develop new GERD symptoms at any point of time after surgery, and thus the incidence is high especially within the first 6 months, but is expected to improve with time. Our study data shows that after 6 months, there was in fact improvement of de novo GERD from 74% to 48.5%. This may be due to the reasons stated above, but we also postulate that eating behavior plays an important role as the patients have a tendency to overeat during the early post-operative period, resulting in increased intragastric pressure and higher likelihood of reflux. Dietary adjustment takes time and subsequently results in improvement or resolution of GERD symptoms.

2) In the methods section, please state the questionnaire that was used to assess GERD preop and postop. Is the GERD questionnaire validated?

Although our unit’s questionnaire is not validated, it was more relevant to post sleeve gastrectomy subjects as it takes into account smoking and alcohol history, responses to acid suppression medication and differentiation of acid reflux versus volume reflux. As mentioned in our discussion, a significant proportion of post-operative GERD symptoms in patients who have undergone laparoscopic sleeve gastrectomy are related to regurgitation and volume reflux. Althuwani et al. had concluded that 35.7% of reflux is in fact non-acid regurgitation. The same questionnaire was used by Lim et al in the study “Resolution of Erosive Esophagitis after Conversion from Vertical Sleeve Gastrectomy to Roux-en-Y Gastric Bypass” (Obesity Surgery, August 2020). They explained that current validated questionnaires like GERD-Q or GERD-HRQL are useful instruments to assess acid reflux but they do not measure symptoms due to volume reflux like postprandial regurgitation.

3) The methods section stated that the patient answered if GERD is resolved, improved, unchanged or worsen. However, the results section in the tables only showed if GERD is present or not in the postoperative period. Please show details of results such as how many patients with continued GERD symptoms but "improved" compared to preop.

Of the 86 patients with GERD symptoms before surgery, 20 patients had resolution of their symptoms. Another 10 patients reported ongoing but improved GERD symptoms. The rest of the patients had no change or worsening of their symptoms after LSG.

---

## [Decision Letter · Decision Letter 1]

22 Oct 2020

Small hiatal hernia and postprandial reflux after vertical sleeve gastrectomy: a multiethnic Asian cohort

PONE-D-20-04688R1

Dear Dr. Lye,

We’re pleased to inform you that your manuscript has been judged scientifically suitable for publication and will be formally accepted for publication once it meets all outstanding technical requirements.

Kind regards,

Chun Chieh Yeh, M.D., Ph.D.

Academic Editor

PLOS ONE

Additional Editor Comments (optional):

After revisions, I consider the current article could be accepted because the revised contents can response to our reviewers' comments specifically. In addition, a kind suggestion: in your rebuttal letter, you should point out specific locations that the revised content are added.

Reviewers' comments:

Reviewer's Responses to Questions

**Comments to the Author**

1. If the authors have adequately addressed your comments raised in a previous round of review and you feel that this manuscript is now acceptable for publication, you may indicate that here to bypass the “Comments to the Author” section, enter your conflict of interest statement in the “Confidential to Editor” section, and submit your "Accept" recommendation.

Reviewer #1: All comments have been addressed

2. Is the manuscript technically sound, and do the data support the conclusions?

Reviewer #1: (No Response)

3. Has the statistical analysis been performed appropriately and rigorously? 

Reviewer #1: (No Response)

4. Have the authors made all data underlying the findings in their manuscript fully available?

Reviewer #1: (No Response)

5. Is the manuscript presented in an intelligible fashion and written in standard English?

Reviewer #1: (No Response)

6. Review Comments to the Author

Reviewer #1: (No Response)

7. PLOS authors have the option to publish the peer review history of their article (what does this mean?). If published, this will include your full peer review and any attached files.

Reviewer #1: No

---

## [Editor Report · Acceptance letter]

28 Oct 2020

PONE-D-20-04688R1 

Small hiatal hernia and postprandial reflux after vertical sleeve gastrectomy: a multiethnic Asian cohort 

Dear Dr. Lye:

I'm pleased to inform you that your manuscript has been deemed suitable for publication in PLOS ONE. Congratulations! Your manuscript is now with our production department. 

Kind regards, 

on behalf of

Dr. Chun Chieh Yeh 

Academic Editor

PLOS ONE